# Adherence to iron-folic acid supplementation and its associated factors among pregnant women in Kenya: A multilevel data analysis of the 2022 Kenyan Demographic and Health Survey

Gebrie Getu Alemu[1]*, Dagnew Getnet Adugna[2], Amare Mesfin[3], Muluken Chanie Agimas[1], Lemlem Daniel Baffa[3], Habtamu Wagnew Abuhay[1], Mekuriaw Nibret Aweke[3], Tewodros Getaneh Alemu[4], Nebebe Demis Baykemagn[5]

1 Department of Epidemiology and Biostatistics, Institute of Public Health, College of Medicine and Health Sciences, University of Gondar, Gondar, Ethiopia, 2 Department of Human Anatomy, School of Medicine, College of Medicine and Health Science, University of Gondar, Gondar, Ethiopia, 3 Department of Human Nutrition, Institute of Public Health, College of Medicine and Health Sciences, University of Gondar, Ethiopia, 4 Department of Pediatrics and Child Health Nursing, School of Nursing, College of Medicine and Health Sciences, University of Gondar, Gondar, Ethiopia, 5 Department of Health Informatics, Institute of Public Health, College of Medicine and Health Sciences, University of Gondar, Gondar, Ethiopia

* gebryegetu27@gmail.com

## Abstract

### Background

Iron and folic acid deficiency is a global public health issue, particularly in low- and middle-income countries. Adherence to iron-folic acid supplementation (IFAS) remains low in Kenya. Despite several studies on IFAS adherence in Kenya, they do not assess the national scope and clustering effects using advanced analytical models. Therefore, we aimed to assess IFAS adherence and its associated factors among pregnant women in Kenya using data from the 2022 Kenyan Demographic and Health Survey.

### Methods

A secondary data analysis was conducted using data from the 2022 Kenya Demographic and Health Survey, which was obtained from the official Demographic and Health Survey program database. Weighted samples of 8,460 participants were used in the analysis. The study employed a multivariable multilevel mixed-effects logistic regression model. Variables from the bi-variable model that had p < 0.25 were considered in the multivariable analysis. Variables with a p-value < 0.05 were considered statistically significant in the multivariable model, and the adjusted Odds Ratio (aOR) with its 95% CI was reported.

**Data availability statement:** All relevant data are within the manuscript.

**Funding:** The author(s) received no specific funding for this work.

**Competing interests:** The authors have declared that no competing interests exist.

**Abbreviations:** AIC, Akaike Information Criterion; ANC, Antenatal Care; aOR, adjusted Odds Ratio; BIC, Bayesian information criterion; CI, Confidence Interval; CSA, Central Statistical Agency; DHS, Demographic and Health Survey; ICC, Intra Class Correlation Coefficient; ICF, Inner-City Fund; IDA, Iron deficiency Anemia; IFAS, Iron Folic Acid Supplementation; Hgb, Hemoglobin; KDHS, Kenyan Demographic and Health Survey; LL, Log-Likelihood; MOR, Median Odds Ratio; NGOs, Non-Governmental Organizations; NTDs, Neural Tube Defects; PCV, Proportional Change in Variance; SDG, Sustainable Development Goal; SSA, sub-Saharan Africa; VIF, Variance Inflation Factor; WHO, World Health Organization

## Results

The prevalence of IFAS among pregnant women in Kenya was 61.72% (95% CI: 60.68, 62.75). Women aged 20–39 years were 1.3 to 1.7 times more likely to adhere to IFAS during pregnancy, with the likelihood increasing steadily by age group. Moreover, primigravida [aOR = 1.22 (95% CI: 1.06, 1.41)], with a family size of ≥ 10 [aOR = 0.72 (95% CI 0.59, 0.90)], antenatal care visits ≥ 4 [aOR = 3.96 (95% CI: 1.91, 8.23)], first start of antenatal care at the third trimester of pregnancy [aOR = 0.30 (95% CI: 0.15, 0.62)], and with a higher level of education [aOR = 1.6 (95% CI: 1.35, 1.90)] were statistically significant factors associated with adherence to IFAS.

## Conclusion and recommendation

Nearly two in five pregnant women didn't receive IFAS for the recommended periods. Factors such as maternal age, parity, antenatal visits, and education enhanced IFAS adherence, whereas large family size and starting antenatal care in the third trimester of pregnancy reduced adherence. Therefore, the community, government, and non-governmental organizations should enhance adherence by implementing customized interventions on the factors identified as positive and negative associations.

## Introduction

Adherence in a medical context refers to how well a patient follows a prescribed treatment plan, which is essential for optimal health outcomes [1,2]. Iron, a key component of hemoglobin, is essential for blood formation and oxygen transport. Iron deficiency, one of the most preventable dietary deficiencies, commonly affects women during pregnancy [3]. Iron-folic acid supplementation coverage is a core process indicator of the Global Nutrition Monitoring Framework (GNMF) among pregnant women [4].

Studies have shown that the risk of megaloblastic anemia in pregnant women is dramatically lowered by folic acid supplementation, usually taken with iron. For example, pregnant women given folic acid and iron rather than iron only had a reduction in megaloblastic anemia from 13% to 5%. Besides, periconception IFAS is effective in reducing the risk of neural tube defects (NTDs) such as spinal bifida and anencephaly. Even if iron is often combined with folic acid in supplements primarily to address anemia, the NTD prevention benefits are attributable to folic acid [5,6].

Poor adherence to IFAS is a critical public health challenge in sub-Saharan Africa (SSA). While adherence to IFAS during pregnancy in high-income countries ranges from 77% in Denmark [7] to 85% in Sweden [8], adherence to IFAS among pregnant women in Kenya has been reported to be as low as 10.6% [9]. A population-based survey performed in Brazil reported that the prevalence of IFAS among pregnant women was 59% [10]. Another study done in India showed that the overall compliance of IFAS was reported to be 64.7% [11]. A large population-based study conducted in 22 SSA countries showed that the overall prevalence of adherence to IFAS

among pregnant women was only 28.7% [12]. A systematic review and meta-analysis in SSA revealed that the estimated pooled prevalence of IFAS among pregnant women was 39.2% [13]. Different studies conducted in Ethiopia showed that the prevalence of adherence to IFAS among pregnant women ranged from 17.1% to 84% [14–18]. A study conducted in Tanzania showed the prevalence of IFAS was 22% [19]. Similarly, the prevalence of adherence to IFAS among pregnant women in Kenya ranged from 10.6% [9] to 60.4% [2].

Iron deficiency is a global public health issue, especially in low- and middle-income countries. Iron's limited bioavailability and high prenatal iron needs, particularly in developing nations, raise concerns about the need for additional iron sources such as supplements. The World Health Organization (WHO) recommends routine iron supplementation for all pregnant women in response to the increased demand [20]. To meet the nutritional needs of the growing fetus and their own, pregnant women need more iron and folic acid. Iron and folic acid deficiencies during pregnancy may have detrimental effects on the mother's health and the development of the fetus. Pregnant women are required to take 400 μg (0.4 mg) of folic acid and 30–60 mg of elemental iron orally every day to prevent maternal anemia, puerperal sepsis, a low birth weight, and premature birth [20]. Taking iron supplements during pregnancy can reduce pregnancy-related complications like low birth weight, preterm birth, and postpartum hemorrhage [17].

Anaemia in pregnancy is defined as a hemoglobin (Hgb) content of less than 11 g/dl [21]. It affects almost two billion people worldwide [22]. Anaemia has been observed in about 37% of pregnant women worldwide, with Africa bearing the brunt of this condition despite the provision of iron-folic acid during their antenatal care [23]. According to a recent systematic review and meta-analysis, 35.6% of pregnant women in SSA suffered from pregnancy-related anaemia [13].

In Kenya, the most current micronutrient survey in the country indicated the prevalence of anaemia due to iron deficiency is thought to affect 55.1% of pregnant women [24]. Pregnancy-related anemia is a public health issue that can result in several potentially fatal consequences as well as unfavorable pregnancy outcomes [25]. One of the major global public health issues, accounting for 1.45% of all disability-adjusted life years, is iron and folic acid deficiency anemia. An excessively high percentage of maternal and perinatal illness and mortality is acknowledged to be caused by it [14].

Iron deficiency anemia (IDA) is the most common type of anemia that occurs during pregnancy and is associated with a wide range of nutritional deficiencies that can have detrimental effects on both the mother and the fetus. Most women's stocks of iron are insufficient to fulfill the sharply increased needs during the second and third trimesters of pregnancy. Due to increased dietary requirements, such as those for iron, folate, and vitamin B12, as well as hemodilution during pregnancy, pregnant women are at risk of developing anemia [26]. Maternal anemia's pathogenesis has been connected to iron deficiency, and it significantly contributes to pregnancy-related morbidity and mortality [2]. It is accompanied by an increased risk of maternal mortality, obstetric problems, preterm delivery, and low birth weight [27].

Numerous studies have stated the relationship between different factors and adherence to IFAS. Age of the mother [12,17], place of residence [1], marital status of the mother [12], level of education [1,2,12,27,28], household wealth index [12,27], number ANC visits [1,12,16,17,27,28], timing of ANC visit [12,14,16], birth intervals (parity) [2,16], family size [17], lack of information about the need for IFAS [29], media exposure [27], and distance to health facility [12,27] were significant predictors of adherence to IFAS.

Supplementing with iron-folic acid is a useful strategy to lower the incidence of anemia during pregnancy [30]. It is critical to follow iron supplements throughout pregnancy to minimize anemia and the morbidities that come with it. Pregnant women who receive and use iron supplements appropriately reduce the risk to their health and the health of their unborn children [14]. Iron supplements help prevent and treat anemia in pregnant and postpartum women, improving maternal and perinatal health [31]. Iron-folic acid supplements should be given to all pregnant mothers regularly [17]. Pregnancy increases dietary needs, making iron supplements essential for health. Pregnant women in Kenya can receive free supplements at public health institutions during prenatal appointments, but uptake remains low [2].

Providing and properly using IFAS during pregnancy can help prevent health risks for both the mother and the baby. Iron-folic acid is essential for pregnant women because it supports the mother's increased blood volume, helps prevent

anemia, and contributes to the baby's development. Adequate supplements can improve maternal health, reduce the risk of complications during pregnancy, and lead to better health outcomes for the baby, both at birth and in the early stages of life. There is no national representative information regarding IFAS among pregnant women in Kenya. Performing a countrywide analysis with recent nationally representative Demographic and Health Survey (DHS) data in Kenya is essential for identifying common determinants in the country. Reducing maternal mortality is a key indicator for achieving the third Sustainable Development Goal (SDG) on good health and wellbeing, aiming to lower maternal deaths to less than 70 per 100,000 live births. Therefore, we aimed to determine the prevalence of iron-folic acid supplementation and its associated factors among pregnant women in Kenya.

## Methods

### Study setting

Kenya is an East African country that shares borders with five other countries: Uganda to the West, South Sudan to the Northwest, Ethiopia to the North, Somalia to the East, and Tanzania to the South. Its Southeast flank abuts the Indian Ocean. The total area of Kenya encompasses approximately 580,650 km$^2$ with a total population of 57,212,424. With an area of 71,597.8 sq. km, Turkana is the largest county, while Nairobi, which is the national capital city, is the most populous one with a population of 4,397,075 [32,33]. Kenya, a low and middle-income African country, is divided into 47 administrative regions, each categorized into rural and urban areas, with 92 strata, since Nairobi City and Mombasa counties are purely urban [34]. Kenya's health system, managed by national and county governments, operates through a six-tier structure, involving community units and tertiary Referral Hospitals. Despite progress, the quality of health services remains insufficient due to poor infrastructure and availability [35].

### Study design

Secondary data analysis was done on Kenya Demographic and Health Survey (KDHS) 2022, which was collected by using a cross-sectional study design.

### Study period

Data used were collected from Kenyan women of reproductive age who are in the ages of 15–49 years, with data collection taking place from February 17 to July 19, 2022, in all counties of the country [36].

### Data source and sampling procedure

The data were accessed from the official database of the DHS program (www.measuredhs.com) after permission was secured through an online request by explaining the purpose of this study. The DHS is a population-based survey that is nationally representative and is gathered using standard questions and equivalent national manuals. The DHS collects a variety of objective and self-reported data with a particular focus on indicators of fertility, reproductive health, mother and child health, mortality, nutrition, and self-reported health characteristics among adults. High response rates, nationwide reach, excellent interviewer training, uniform data collection practices across nations, and constant content over time are some of the DHS's main advantages. The KDHS 2022 was used to draw this study data. In Kenya, the 2022 KDHS is the seventh survey of its kind. The 2022 KDHS samples were selected using a two-stage stratified cluster sampling technique to select study participants based on the 2019 Kenya Population and Housing Census (KPHC) as a sampling frame. In the first stage, 1,692 clusters were selected from the Kenya Household Master Sample Frame (K-HMSF) using the Equal Probability Systematic Sampling Method (EPSSM). In every sampling stratum, the clusters were selected independently. All the clusters that were chosen had their homes listed, and the resulting list of households was used from each cluster. All the households from these clusters were chosen for the sample because, following the household listing method, it

was discovered that certain clusters contained less than 25 households. The 2022 KDHS sample was designed to be a reliable estimate at the national level, for rural and urban areas individually, and, for some variables, at the county level for each of the forty-seven counties, which comprised 42,022 households [34]. Interviews were conducted face-to-face using tablet computers with computer assisted personal interviewing technology. Interviews were available for all women aged 15–49 who regularly resided in the chosen houses or who spent the night before the survey there. The 2022 KDHS interviewed a nationally representative sample of 32,156 women aged 15–49; of these, 11,863 women had recently given birth within the five years preceding the survey [34]. Among 11,863 women who had recently given birth within the five years preceding the survey, 9,312 women had at least one ANC follow-up. Among those women who had at least one ANC follow-up (9,312), 344 were excluded because they did not take IFAS and for incompleteness. Finally, 8,969 (8,460 weighted) samples were included in the final analysis, and it was inquired of them how many days they had taken iron tablets or syrups during their previous pregnancy (Fig 1).

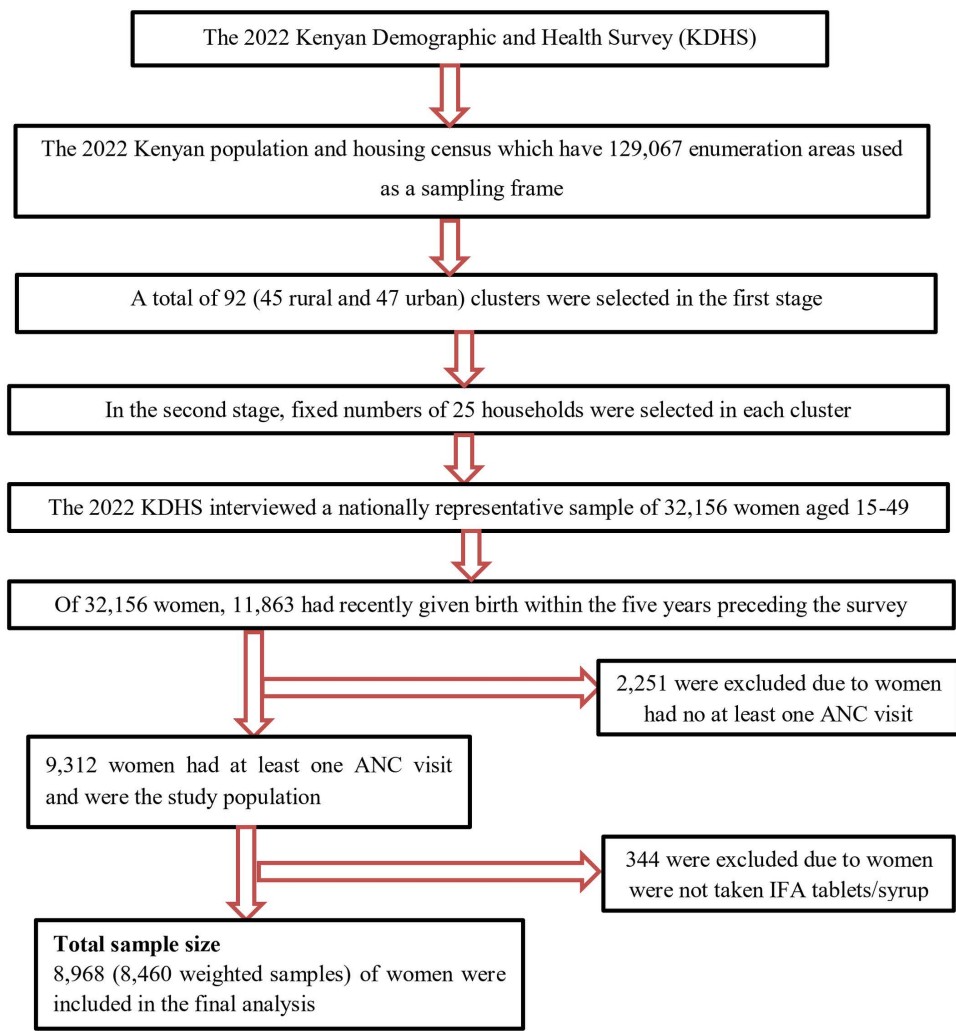

**Fig 1. Flow diagram of the 2022 KDHS sampling design and data extraction process.**

## Source and study population

**Source population.** All Kenyan women aged 15–49 years, who had a live birth in the five-year preceding 2022 KDHS. Out of 32,156 women who responded the questionnaire, 11,863 had given birth in the five years before the 2022 KDHS [34].

**Study population.** The study population consisted of Kenyan women aged 15–49 who had a live birth between 2017 and 2022, five years preceding the 2022 KDHS. Women who attended at least one ANC visit (9,312) were included and only information pertaining to the most recent pregnancy were used.

## Ethical approval and consent to participate

The Institutional Review Board (IRB) of the Inner-City Fund (ICF) granted ethical approval for the 2022 KDHS. The Kenya National Bureau of Statistics conducted the survey in collaboration with other development partners. Written informed consent was obtained from all participants or the legally appointed representative of minor participants. Respondents participated voluntarily without coercion [37], and all data are Health Insurance Portability and Accountability Act (HIPAA) protected and de-identified. These ethical issues were addressed by those who conducted the primary surveys, not the current authors of the manuscript. Since this study employed already-existing public domain survey data sets with all identifiers' information removed that are freely accessible online at www.measuredhs.com, ethical approval was not necessary because it was taken with ICF. However, we had an authorization letter from Measure DHS to access and utilize the data by submitting an online request.

## Measurement of variables

**Dependent variable.** The outcome variable in this study was adherence to iron-folic supplementation during recent pregnancy. Adherence to IFAS was defined as taking iron tablets or syrups for ≥ 90 days during antenatal care. Adherence to IFAS in the DHS data was measured by the number of days iron tablets or syrups were taken during antenatal care, with ≥ 90 days selected based on WHO guidelines [38].

**Independent variables.** Individual and community-level variables were considered independent variables based on the DHS data and literature review (S1 Table).

**Individual level variables:** maternal age, marital status, education, occupation, wealth, history of pregnancy termination, ANC attendance and timing, household head's sex, birth interval, family size and perceived distance to the health facility.

**Community-level independent variables:** Previous studies used mean or median values to aggregate individual-level characteristics into community variables [39,40].

Place of residence (urban, rural) was originally recorded, so aggregation was not needed.

**Media exposure** was measured by the percentage of women exposed to at least one form of media, such as television, radio, or newspapers. Participants exposed to at least one of the three media were labeled "exposed" and coded as "1"; otherwise, "not exposed" was coded as "0".

**Community education:** This aggregate variable represents the percentage of women in the community with low or high education. Values below the median are classified as low, while those at or above the median are classified as high.

**Community wealth index:** Women were categorized by the community wealth index based on their households as living in poor or rich households. The percentage of women in each category was calculated, and communities were classified as "low" (< 45.58%) or "high" (≥ 45.58%) using the median percentage as the cutoff.

Community-level women's education, media exposure, and wealth index were created by aggregating individual-level data, as these variables were not directly available in the DHS. Since they were not normally distributed, the median was used to categorize communities as low or high based on the proportion of women's education, media exposure, and wealth index.

 

## Statistical analysis

STATA 17 was used for data extraction, recoding, and analysis. Weighting was applied to enhance sample representativeness, ensure population resemblance, and obtain reliable estimates. The hierarchical structure of KDHS data violates the classical logistic regression model's assumptions of independent observations and equal variance. Pregnant women are nested within a cluster, and we expect that pregnant women within the same cluster may be more similar each other than pregnant women found in different clusters. This highlights the need to use a multilevel mixed-effect logistic regression model to account for variability between clusters. Therefore, a multilevel mixed-effect logistic regression model was fitted to estimate the association between the individual and community-level variables and adherence to iron-folic acid supplementation.

Cluster variation was quantified using the Intra Class Correlation Coefficient (ICC), Median Odds Ratio (MOR), and Proportional Change in Variance (PCV). The ICC measured the similarity of individuals within a cluster, distinguishing it from cluster heterogeneity [41]. In this study, ICC was used for understanding how much of the adherence variability is explained by cluster differences rather than individual differences. The intra-class correlation was calculated as the proportion of the between-community variation in the total variation alive.

$$ICC = \frac{\sigma^2}{\sigma^2 + \tau_0^2}$$

(1)

where, $\sigma^2$ is the cluster level variance and $\tau_0^2$ is the individual level variance.

The ICC/Rho from the empty model tells how much of the total variance in the adherence to iron supplementation is accounted for by the clustering or grouping structure of the data. A high ICC (closer to 1) indicates substantial clustering; meaning individuals within the same group are more similar to each other than to individuals from different groups. This suggests the need to account for the clustered nature of the data using a multilevel model. A low ICC (closer to 0) indicates little clustering, meaning there is more variation within groups than between groups. This suggests the clustered structure may not be important, and a single-level model may be sufficient.

When we select two random clusters, the MOR is the median value of the odds ratios between the areas with the highest and lowest risk. It can be understood as when someone moves from a low-risk cluster to a higher-risk cluster, they are exposed to a greater degree of risk. In this study, the MOR shows the individual increased risk of adherence to IFAS when someone moves from cluster with low probability of adherence to IFAS to area with high risk of adherence to IFAS. The MOR is always greater than or equal to 1. The likelihood of adherence to IFAS would not vary among clusters if MOR was equal to 1 [42].

$$MOR = \exp(\sqrt{2 * \sigma^2 * 0.6745}) = \exp(0.95\sqrt{\sigma^2})$$

(2)

where; 0.6745 is the 75th centile of the cumulative distribution function of the normal distribution with mean 0 and variance of 1, and $\sigma^2$ is the cluster level variance.

Proportional (percentage) change in variance is a measure of total variation attributed to individual-level factors and community-level factors in the multilevel model as compared to the null model [42].

$$PCV = \frac{\sigma^2null - \sigma^2var}{\sigma^2null}$$

(3)

where; $\sigma^2null$ is variance of the initial model without any explanatory variable (empty model) and $\sigma^2var$ is the variance of the model with determinant factors. I.e. model two with individual level determinant factors, model three with community level factors and model four (full model) with both individual and community level determinant factors alive.

For the multilevel mixed-effect logistic regression analysis, four models were constructed. The first model was the empty model without determinant factors to assess cluster variation in IFAS adherence. The second model includes individual-level explanatory variables and adherence to IFAS; the third model includes community-level explanatory variables and adherence to IFAS, whereas the fourth model includes both individual- and community-level explanatory variables at the same time and adherence to IFAS.

Akaike's Information Criterion (AIC), Bayesian information criterion (BIC) and Log-Likelihood (LL) were used to compare and select the best-fitted model and a model with the lowest AIC, BIC and the largest LL was considered the best-fitted model (Model-IV). Multicollinearity among the independent variables was checked by Variance Inflation Factor (VIF) and tolerance. As a result, there was no multicollinearity because VIF of each independent variable were < 7. Variables from the bi-variable model that had a p-value less than 0.25 were considered in the multivariable analysis. Finally, aOR with a 95% Confidence Interval (CI) was reported, and variables with p-value < 0.05 in the multivariable multilevel mixed-effect logistic regression were identified as significant predictors of adherence to iron-folic acid supplementation.

## Results

### Characteristics of the study participants

A weighted sample of 8,460 pregnant women aged 15–49 was included in the study. Nearly one-third (29.34%) were aged 25–29, more than two-thirds (70.34%) were from male-headed households, and about 79.14% were married/living with a partner. Regarding the occupational status of the pregnant women, about 43.32% were unemployed. One-third (29.73%) of the study participants perceived distance to the health facility as a big problem. Only about one-third (32.03%) of the survey respondents had received four or more ANC visits, and most (59.95%) began ANC in the second trimester. Additionally, the majority (72.55%) of women were multigravida. Furthermore, more than two-thirds (70.29%) of the study participants had a prenatal nurse/midwife at the most recent birth. Regarding the community-level characteristics, the majorities (62.17%) of the study participants were rural dwellers, and about 44.94% were from literate communities. Finally, about 61.68% of the study participants were from a high level of community wealth, whereas about 60.60% had community media exposure (Table 1).

The prevalence of iron-folic acid supplementation among pregnant women in Kenya was 61.72% (95% CI: 60.68, 62.75) (Fig 2).

### Factors associated with adherence to iron-folic acid supplementation among pregnant women in Kenya

A multilevel mixed-effects binary logistic regression model was used to identify potential factors affecting IFAS adherence. The final model (model-IV) was deemed suitable for identifying individual and community-level variables associated with IFAS adherence in Kenya, following an evaluation of model fitness using various post-estimation methods, including AIC, BIC, and LL. The variability in adherence to IFAS among pregnant women attributed to cluster-level variability was 15.77% (95% CI: 13.32, 18.58) based on the estimated ICC, while 84.33% was attributed to individual-level factors (Table 2).

Fifteen independent variables with $p < 0.25$ in the bivariable analysis were included in the multivariable multilevel mixed-effects logistic regression. At a 95% confidence level, the significant factors associated with IFAS identified from the best-fitted model (model-IV) were maternal age, parity, family size, number of ANC visits, timing of first ANC visit, and educational level of the mother.

Being in the age group from 20–24 years was 1.32 times more likely to adhere to iron-folic acid as compared with the age group from 15–19 years [aOR = 1.32 (95% CI: 1.07, 1.63)]. Moreover, women aged 25–29 years were 1.47 times [aOR = 1.47 (95% CI: 1.17, 1.85)] more likely to have adherence to iron-folic acid as compared to women aged 15–19 years. Furthermore, women aged 30–34 years were 1.56 times [aOR = 1.56 (95% CI: 1.21, 2.01)] more likely to have adherence to iron-folic acid as compared to women aged 15–19 years. Being in the age group from 35–39 years was 1.67 times more likely to adhere to IFAS as compared with those aged 15–19 years [aOR = 1.67 (95% CI: 1.27, 2.19)].

**Table 1. Individual and community level weighted socio-demographic characteristics of the pregnant women in Kenya, KDHS 2022.**

| Characteristics (variables) | Category | Weighted frequency (N = 8460) | Weighted percentage (%) |
|---|---|---|---|
| Maternal age (Years) | 15-19 | 589 | 6.96 |
| | 20-24 | 2,270 | 26.83 |
| | 25-29 | 2,482 | 29.34 |
| | 30-34 | 1,641 | 19.40 |
| | 35-39 | 1,033 | 12.20 |
| | 40-49 | 445 | 5.27 |
| Sex of the household head | Male | 5,990 | 70.80 |
| | Female | 2,470 | 29.20 |
| Marital status | Never married | 1,025 | 12.12 |
| | Married/living with partner | 6,696 | 79.14 |
| | widowed/divorced/separated | 739 | 8.74 |
| Maternal occupation | Not working | 3,665 | 43.32 |
| | Working | 4,795 | 56.68 |
| Perception of the distance to health facility | A big problem | 5,945 | 70.27 |
| | Not a big problem | 2,515 | 29.73 |
| Currently pregnant | No | 8,020 | 94.80 |
| | Yes | 440 | 5.20 |
| Parity | Primipara | 2322 | 27.45 |
| | Multipara | 6138 | 72.55 |
| Family size | 1-4 | 3,502 | 41.40 |
| | 5-9 | 4,471 | 52.85 |
| | 10+ | 487 | 5.75 |
| ANC visit | 1-3 | 5,750 | 67.97 |
| | ≥4 | 2,710 | 32.03 |
| Timing of first ANC visit | First trimester | 2,685 | 31.74 |
| | Second trimester | 5,072 | 59.95 |
| | Third trimester | 703 | 8.30 |
| Prenatal nurse/midwife | No | 2,514 | 29.71 |
| | Yes | 5,946 | 70.29 |
| Residence | Urban | 3,200 | 37.83 |
| | Rural | 5,260 | 62.17 |
| Community literacy | Low | 4658 | 55.06 |
| | High | 3802 | 44.94 |
| Community wealth | Low | 5,218 | 61.68 |
| | High | 3,242 | 38.32 |
| Community media exposure | Low | 3,333 | 39.40 |
| | High | 5,127 | 60.60 |

Parity is one of the determinant factors associated with IFAS. The odds of adherence to IFAS were 1.22 times [aOR = 1.22 (95% CI: 1.06, 1.41)] higher among primipara women compared with multipara. Those with larger families were less likely to take iron-folic acid during pregnancy. The odds of adherence to IFAS among pregnant mothers with a family size of ≥ 10 were reduced by 28% [aOR = 0.72 (95% CI 0.59, 0.90)] compared to those with a family size of 1–4.

In terms of ANC, the odds of adherence to IFAS were nearly four times [aOR = 3.96 (95% CI: 1.91, 8.23)] higher for mothers who attended the minimum four ANC visits compared to those who didn't attend the minimum four ANC

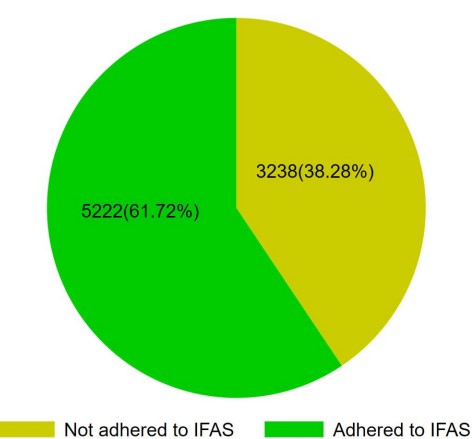

██ Not adhered to IFAS    ██ Adhered to IFAS

**Fig 2. Adherence level of iron-folic acid supplementation among pregnant women in Kenya, KDHS 2022.**

**Table 2. Random effect analysis and model comparison for adherence to IFAS among pregnant women in Kenya, KDHS 2022.**

| Random effect | Model-I (Empty) | Model-II | Model-III | Model-IV |
|---|---|---|---|---|
| ICC % | 15.77 (13.32, 18.58) | 14.32 (11.84, 17.22) | 14.46 (12.09, 17.22) | 13.76 (11.31, 16.63) |
| MOR | 2.1 | 2.04 | 2.03 | 1.99 |
| PCV | Ref. | 0.091 | 0.097 | 0.144 |
| Model Fitness | | | | |
| Log Likelihood | −5923.3385 | −5524.0175 | −5883.029 | −5506.3737 |
| AIC | 11850.68 | 11092.03 | 11778.06 | 11064.75 |
| BIC | 11864.88 | 11249.25 | 11820.67 | 11248.36 |

AIC: Akaike Information Criterion; BIC: Bayesian information criterion; ICC: Intra Class Correlation Coefficient; Ref: Reference; MOR: Median Odds Ratio; PCV: Proportional Change in Variance.

follow-ups. The odds of adherence to IFAS were decreased by 70% [aOR = 0.30 (95% CI: 0.15, 0.62)] among mothers who first started iron-folic acid at the third trimester of pregnancy compared to those who began iron-folic acid at the first trimester of pregnancy. Finally, pregnant women with a higher level of education were 1.6 times more likely to follow IFAS compared to those with low levels of education [aOR = 1.63 (95% CI: 1.35, 1.90)] (Table 3).

## Discussion

The prevalence of IFAS among pregnant women in Kenya was 61.72% (95% CI: 60.68, 62.75). Maternal age, parity, antenatal visits, education, family size, and starting time of ANC were factors significantly associated with adherence to IFAS.

This study found that only 61.72% of mothers in Kenya took IFAS according to WHO recommendations. This study is consistent with studies conducted in Kenya (60.6%) [2], and Ethiopia (60.9%) [43]. The reason could be that since the study results are consistent with findings in Ethiopia and other Kenyan studies, it suggests a broader trend across similar regions. Moreover, these countries have implemented national policies promoting IFAS during pregnancy as part of ANC programs. Furthermore, Kenya and Ethiopia experience overlapping public health concerns, such as malnutrition and maternal health challenges, leading to similarities in medical and epidemiological studies.

The prevalence of IFAS in this study exceeded rates reported in Kenya (10.4%) [9], (32.7%) [44], Ethiopia (18.97%, – 56.5%) [1,12,14–16,27,45], Tanzania (20.3%) [46], Ghana (58.8%) [47], SSA (28.7%) [12] and Brazil (59%) [10]. The discrepancy could stem from study design, because most individual studies focus on institution-level ANC follow-ups.

**Table 3. Multi-level logistic regression analysis of individual and community level factors associated with IFAS among pregnant women in Kenya, KDHS 2022.**

| Characteristics | Category | aOR (95% CI) | | | | |
|---|---|---|---|---|---|---|
| | | Model I (empty) | Model II | Model III | Model IV | P-value |
| Maternal age (Years) | 15-19 | | 1 | | 1 | |
| | 20-24 | | 1.34 (1.09, 1.66) ** | | 1.32 (1.07, 1.63) * | 0.010 |
| | 25-29 | | 1.47 (1.17, 1.85) ** | | 1.47 (1.17, 1.85) ** | 0.002 |
| | 30-34 | | 1.47 (1.15, 1.88) ** | | 1.56 (1.21, 2.01) ** | 0.002 |
| | 35-39 | | 1.51 (1.16, 1.96) ** | | 1.67 (1.27, 2.19) *** | < 0.001 |
| | 40-49 | | 1.39 (0.94, 1.75) | | 1.48 (0.96, 2.07) | 0.163 |
| Sex of the household head | Male | | 1 | | 1 | |
| | Female | | 1.05 (0.93, 1.19) | | 1.08 (0.96, 1.21) | 0.211 |
| Marital status | Never married | | 1 | | 1 | |
| | Married | | 0.99 (0.82, 1.20) | | 1.05 (0.87, 1.27) | 0.664 |
| | Separated | | 0.85 (0.67, 1.07) | | 0.88 (0.69, 1.12) | 0.291 |
| Maternal occupation | Not working | | 1.00 (0.90, 1.11) | | 1.06 (0.96, 1.19) | 0.248 |
| | Working | | 1 | | 1 | |
| Perception of the distance to HF | Not a big problem | | 1 | | 1 | |
| | A big problem | | 0.95 (0.86, 1.06) | | 0.96 (0.87, 1.07) | 0.425 |
| Currently pregnant | No | | 1 | | 1 | |
| | Yes | | 0.97 (0.79, 1.19) | | 1.01 (0.82, 1.23) | 0.938 |
| Parity | Primipara | | 1.27 (1.09, 1.46) ** | | 1.22 (1.06, 1.41) ** | 0.007 |
| | Multipara | | 1 | | 1 | |
| Family size | 1-4 | | 1 | | 1 | |
| | 5-9 | | 0.88 (0.78, 0.99) | | 0.89 (0.79, 1.00) | 0.056 |
| | 10+ | | 0.69 (0.56, 0.85) *** | | 0.72 (0.59, 0.90) ** | 0.001 |
| ANC visit | 1-3 | | 1 | | 1 | |
| | ≥4 | | 3.48 (1.68, 7.22) *** | | 3.96 (1.91, 8.23) *** | < 0.001 |
| Timing of first ANC visit | First trimester | | 1 | | 1 | |
| | Second trimester | | 1.79 (0.86, 3.75) | | 2.05 (0.98, 4.29) | 0.061 |
| | Third trimester | | 0.27 (0.13, 0.54) *** | | 0.30 (0.15, 0.62) *** | < 0.001 |
| Prenatal nurse/midwife | No | | 1 | | 1 | |
| | Yes | | 1.05 (0.94, 1.18) | | 1.08 (0.96, 1.21) | 0.211 |
| Residence | Urban | | | 1 | 1 | |
| | Rural | | | 0.94 (0.82, 1.08) | 0.98 (0.85, 1.13) | 0.806 |
| Community literacy | Low | | | 1 | 1 | |
| | High | | | 1.63 (1.39, 1.91) *** | 1.60 (1.35, 1.90) *** | < 0.001 |
| Community poverty | Low | | | 1 | 1 | |
| | High | | | 0.81 (0.72, 0.92) | 0.93 (0.81, 1.06) | 0.174 |
| Community media exposure | Low | | | | 1 | |
| | High | | | 1.03 (0.89, 1.17) | 0.97 (0.84, 1.10) | 0.661 |

aOR; adjusted Odds Ratio, CI; Confidence Interval, 1; Reference category, ANC; Antenatal care, HF; Health Facility, *P<0.05, **P<0.01, ***P<0.001.
***Note***: The included P-values were for Model IV.

In contrast, this study employed a large-scale survey, and others relied on meta-analysis [48]. Moreover, the observed difference could be attributed to study periods, sample size, study setting, and sociodemographic characteristics [49,50]. Furthermore, enhanced health-seeking behavior and increased ANC visits could have contributed to higher adherence [36,48].

This study's finding was lower than those reported in Ethiopia (84%) [30], Niger (68.6%) [51], India (71%) [52], Denmark (77%) [7] and Sweden (85%) [8]. The difference could be attributed to the intermittent nature of DHS data collection (once every 5 years), leading to potential recall bias and underestimation of IFAS adherence, lack of knowledge of IFAS, and absence of supply chain information [48]. Moreover, the lower IFAS adherence in Kenya compared to others could be due to a combination of supply chain issues, inadequate counseling, lack of knowledge and information about maternal anemia and the benefits of IFAS among both pregnant women and health care providers, broader socio-economic barriers, and inaccessibility to ANC services, especially in rural and marginalized areas [2,53].

In this study, older women showed higher IFAS compliance than younger women, with those aged 20–24, 25–29, 30–34 and 35–39 years were more likely adhered compared to those aged 15–19. Our findings aligned with studies conducted in Ethiopia [17,54], Tanzania [19] Sudan [55], SSA [12], India [56] and Brazil [57]. The correlation between IFAS adherence and age may be explained by older women's knowledge of the advantages of iron supplementation benefits or their personal experiences with iron deficiency. Older women may be more prone to anemia due to recurrent pregnancies and relay on iron supplements prevention. They may also be more health-conscious, receive family support, and have greater experience in managing iron deficiency anemia [10]. Pregnant adolescents often feel shy, and in this community, their pregnancies may be perceived as unusual. Unplanned pregnancies further worsen the situation, leading to stress and reducing their likelihood of seeking ANC services. Discomfort with medical professionals also discourages ANC visits, and decreasing motivation [11]. This finding highlights the Ministry of Health (MoH) to equip healthcare workers with the necessary skills and knowledge to support these mothers and facilitate their cares.

Pregnant women from families with ≥ 10 members were 28% less likely to adhere to IFAS than those from smaller families (1–4 members), aligning with findings from a study in Ethiopia [17]. The reason may be because mothers with larger families attend ANC less frequently due to household responsibilities and caregiving duties [19]. Large family size often means limited financial and material resources could be shared among more individuals, which can reduce the priority given to maternal health needs, such as IFAS adherence. In such households, pregnant women may face increased domestic and caregiving responsibilities, leading to time constraints and forgetfulness, which are recognized barriers to supplement adherence. In large families, individual health support may be weaker, as attention and encouragement are key to IFAS compliance, and can be spread when there are many dependents [58,59].

Parity is a significant factor in IFAS adherence. Primiparous mothers are 1.22 times more likely to comply than multipara respondents, aligning with studies in Kenya [2,44] and India [60]. The reason could be that primipara mothers are more likely to adhere to IFAS due to increased motivation, better engagement with ANC, higher perceived risk, and fewer competing responsibilities [61]. However, this findings contrasts with studies in Ethiopia, where multipara mothers were more likely to adhere to IFAS than primipara mothers [61]. The reason could be that women with positive birth histories or severe iron-folic acid side effects may not fully recognize the importance of continued supplementation in later pregnancies [29]. This implies that regardless of prior experiences, educating multigravida mothers on the importance of IFAS adherence in subsequent pregnancies is essential.

Mothers who attended the minimum recommended ANC visits had higher iron-folic acid compliance. Mothers who attended at least four ANC visits had nearly four times higher odds of IFAS adherence than those who did not. This finding is consistent with studies conducted in Kenya [2], Ethiopia [62,63], and SSA [12]. The reason could be that more ANC visits increase contact between mothers and healthcare professionals, providing opportunities for supplement counseling and supply replenishment. These visits also allow mothers to discuss challenges and receive guidance. Encouraging pregnant women to complete the WHO-recommended eight visits can enhance IFAS compliance [62,63].

The time of the first ANC visit significantly affects IFAS adherence. Mothers who began ANC in the third trimester were 70% less likely to adhere than those who started in the first trimester, consistent with studies in Ethiopia [16,64,65]. The reason could be that early ANC visits improve IFAS adherence because supplementation should begin as soon as possible; however, many women delay IFAS initiation by realizing their pregnancy only after the first trimester. Early IFAS is recommended, and adherence is strongly influenced by the timing of the first ANC visit [66]. This highlights the necessity of customized interventions to consider the early commencement and difficulties faced by pregnant women to guarantee optimal adherence to IFAS throughout pregnancy.

Higher education levels were positively associated with IFAS adherence, with educated pregnant women nearly twice as likely to use iron-folic acid supplements compared to those with lower education. This finding is similar to studies conducted in Kenya [2], Ethiopia [1,12,27], Senegal [67], SSA [12], India [68] and Pakistan [69]. The reason could be that education plays a vital role in informing expectant mothers about iron deficiency and its management. This suggests that pregnant women with higher education levels are more likely to utilize maternal health services, including iron supplements. Education is essential in increasing pregnant women's knowledge of iron deficiency and guiding them on how to manage it. Pregnant women with higher education levels are better equipped to benefit from maternal health treatments like iron-folate supplements. They are more informed about pregnancy risks and have greater access to knowledge on the benefits of iron supplementation through various resources [12,43,67].

### Strengths and limitations of the study

One of the strengths of this study is the use of nationally representative weighted data, which allows for generalizability at the national level. A multilevel analysis accounts for hierarchical data structure (individual and community), providing more accurate results. The findings will help to address a major public health problem in LMICs, where iron deficiency anemia contributes to maternal and child morbidity and mortality. The findings will also help to improve counseling and interventions for neural tube defects, megaloblastic anemia, and iron deficiency anemia. It will directly support SDG 3 (Good Health and Well-Being) by improving maternal and child health outcomes. We found suboptimal IFAS adherence among pregnant women in Kenya when compared with WHO guidelines, indicating a need for immediate and targeted interventions. Legislators and decision-makers can use these insights to optimize ANC services and support adherence within the existing health systems. This study also offers baseline data to inform healthcare providers, managers, and NGOs to implement an evidence-based intervention in pregnant women concerning IFAS adherence. Strengthening ANC services, health education, and adherence support is important for reducing anemia, obstetric complications, preterm birth, low birth weight, maternal and perinatal mortality. In summary, this study goes beyond standard DHS reports by employing advanced analytical techniques to uncover context-specific determinants of IFAS adherence in Kenya, offering actionable insights for policymakers and program implementers.

Potential recall bias from self-reported data collected over the preceding five years is one of the study's limitations. However, this bias was reduced by using data from the most recent pregnancy, and it was further minimized by using standardized DHS questionnaires. Additionally, there are inherent limitations due to the data's cross-sectional nature. Due to the use of secondary sources, certain predictors were not available in the 2022 KDHS data, including cultural factors, household activities, and maternal decision-making. Further studies should use qualitative inquiry to explore how sociocultural and economic factors affect Kenyan pregnant women's adherence to IFAS. This method will provide more thorough and accurate recommendations for enhancing adherence by considering the specific barriers and facilitators.

### Conclusion and recommendation

Iron-folic acid supplementation adherence among pregnant women in Kenya was moderate. Nearly two out of every five pregnant women didn't receive IFAS for the recommended periods. Maternal age, parity, family size, education, ANC visits and timing were key factors influencing IFAS adherence. Training should be provided to frontline healthcare professionals

at all levels to enhance IFAS compliance in the workplace. The MoH should improve maternal health services by emphasizing younger women (15–19 years) and women with 10 or more families, as they are less likely to adhere to IFAS. Researchers should also do further qualitative studies that concentrate on other areas of IFAS adherence barriers.

## Supporting information

**S1 Table. Lists of individual and community level factors affecting adherence to iron-folic acid supplementation among pregnant women in Kenya, KDHS 2022.**
(DOCX)

## Acknowledgments

We would like to acknowledge the MEASURE DHS project for their support and free access to the 2022 KDHS original dataset.

## Author contributions

**Conceptualization:** Gebrie Getu Alemu, Muluken Chanie Agimas, Habtamu Wagnew Abuhay, Mekuriaw Nibret Aweke, Tewodros Getaneh Alemu.

**Data curation:** Gebrie Getu Alemu, Dagnew Getnet Adugna, Amare Mesfin, Lemlem Daniel Baffa, Nebebe Demis Baykemagn.

**Formal analysis:** Gebrie Getu Alemu, Mekuriaw Nibret Aweke, Nebebe Demis Baykemagn.

**Funding acquisition:** Muluken Chanie Agimas.

**Investigation:** Gebrie Getu Alemu, Dagnew Getnet Adugna, Amare Mesfin, Habtamu Wagnew Abuhay.

**Methodology:** Gebrie Getu Alemu, Dagnew Getnet Adugna, Muluken Chanie Agimas, Lemlem Daniel Baffa, Mekuriaw Nibret Aweke.

**Project administration:** Gebrie Getu Alemu, Habtamu Wagnew Abuhay, Tewodros Getaneh Alemu, Nebebe Demis Baykemagn.

**Resources:** Gebrie Getu Alemu, Amare Mesfin, Mekuriaw Nibret Aweke.

**Software:** Gebrie Getu Alemu, Dagnew Getnet Adugna, Tewodros Getaneh Alemu.

**Supervision:** Gebrie Getu Alemu, Habtamu Wagnew Abuhay.

**Validation:** Gebrie Getu Alemu.

**Visualization:** Gebrie Getu Alemu, Amare Mesfin, Nebebe Demis Baykemagn.

**Writing – original draft:** Gebrie Getu Alemu, Muluken Chanie Agimas, Mekuriaw Nibret Aweke, Nebebe Demis Baykemagn.

**Writing – review & editing:** Gebrie Getu Alemu, Amare Mesfin, Lemlem Daniel Baffa, Habtamu Wagnew Abuhay, Mekuriaw Nibret Aweke, Tewodros Getaneh Alemu, Nebebe Demis Baykemagn.

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
