## [Decision Letter · Decision Letter 0]

PONE-D-24-57968Adherence to iron-folic acid supplementation and its associated factors among pregnant women in Kenya: a multilevel data analysis of 2022 Kenyan national population based surveyPLOS ONE

Dear Dr. Alemu,

Thank you for submitting your manuscript to PLOS ONE. After careful consideration, we feel that it has merit but does not fully meet PLOS ONE’s publication criteria as it currently stands. Therefore, we invite you to submit a revised version of the manuscript that addresses the points raised during the review process.

Please rework on all your section taking reference with the published paper in PLOS one. Please respond to all the concern of the peer reviewer. Special attention should be given for formatting of Table and Figures along with English. 

We look forward to receiving your revised manuscript.

Kind regards,

Kanchan Thapa, MPH, MPhil

Academic Editor

PLOS ONE

Journal Requirements:

3. We note that your Data Availability Statement is currently as follows: “All relevant data are within the manuscript and in Supporting Information files.”

4. We note you have included a table to which you do not refer in the text of your manuscript. Please ensure that you refer to Table 2 in your text; if accepted, production will need this reference to link the reader to the Table.

Additional Editor Comments:

Dear Authors,

Remove reference in Abstract.

Pleas reorganize the paper based on reviewer comments. Please work on Abstract, Background, Methods, Results, Discussion and Conclusion section. Follow standard for reporting aOR.

Figure and Table should be formatted based on PLOS One guideline and scientific standards.

Provide a letter of granting permission for data usage for analysis as supplementary file.

Reviewers' comments:

Reviewer's Responses to Questions

**Comments to the Author**

1. Is the manuscript technically sound, and do the data support the conclusions?

Reviewer #1: Yes

Reviewer #2: Yes

Reviewer #3: Yes

2. Has the statistical analysis been performed appropriately and rigorously? 

Reviewer #1: Yes

Reviewer #2: Yes

Reviewer #3: Yes

3. Have the authors made all data underlying the findings in their manuscript fully available?

Reviewer #1: No

Reviewer #2: Yes

Reviewer #3: Yes

4. Is the manuscript presented in an intelligible fashion and written in standard English?

Reviewer #1: Yes

Reviewer #2: Yes

Reviewer #3: Yes

5. Review Comments to the Author

Reviewer #1: This study provides relevant and high scientific value information on maternal health, focusing on sociodemographic factors that could be associated with adherence to iron and folic acid supplementation among pregnant women in Kenya. Its relevance in the context of public health is indisputable, given the direct impact these micronutrients have on maternal-fetal health.

Abstract:

The authors are advised to remove the abbreviation “KDHS 2022” from the keywords section, as its inclusion does not significantly contribute to the indexing or visibility of the article and may be redundant.

Introduction:

The article adequately emphasizes the importance of investigating the factors influencing adherence to iron and folic acid supplementation, highlighting it as a critical public health issue with substantial implications for maternal-fetal well-being. However, it is suggested that the authors structure a clear research question at the end of the introduction, aligned with the study's stated objective. This will enhance the clarity and direction of the study.

Method:

The methodological design is appropriate and aligned with the study's objective, supporting the validity of the results. Furthermore, the authors have provided an adequate operational definition of the independent variables, contributing to the reproducibility and understanding of the approach employed.

Implications for Policy, Practice, and Future Research:

The authors are encouraged to restructure the section dedicated to the study's implications, developing and justifying each aspect in separate paragraphs. Specifically:

1. Implications for Public Health Policy: It is recommended to draft a specific paragraph addressing how the findings can inform the design or implementation of policies that promote greater adherence to iron and folic acid supplementation in contexts similar to Kenya.

2. Implications for Healthcare Practice: A second paragraph could focus on practical recommendations for healthcare providers, highlighting strategies to improve education, access, and adherence among pregnant women.

3. Implications for Future Research: A third paragraph should detail priority areas that require further exploration. This section must emphasize the knowledge gaps identified in the study and their relevance for optimizing adherence to iron and folic acid supplementation, with an emphasis on the sociocultural and economic context of Kenya.

Together, these adjustments will strengthen the study’s contribution, enhancing its clarity and utility for both the scientific community and decision-makers in the field of public health.

Reviewer #2: Overall comment

This study is well-conducted, and the researchers are commended for their work thorough evaluating magnitude of Adherence to iron-folic acid supplementation and its associated factors among pregnant women in Kenya. The study is effectively structured, with a clear identification of the impact of Adherence to iron-folic acid supplementation with well thought recommendations for public health interventions. However, to enhance its credibility and ensure clear communication of the findings, the researchers should address the minor comments and suggestions provided above. Once these revisions are made, the study will be ready for publication and will provide valuable insights into malaria prevention and control.

Reviewer #3: Major Concerns:

1. Repetition of contents in introduction:

• The discussion on iron deficiency and its impact on pregnancy is repeated in different paragraphs (e.g., lines 68–74, 95–103). These can be merged into one concise section.

• The importance of iron-folic acid supplementation is reiterated in multiple places (lines 75–83 and 110–119). A single well-structured paragraph would be more effective.

• The risk factors influencing adherence to IFAS (lines 104–109) could be more succinctly stated. Avoid listing too many studies separately and instead synthesize the findings.

• The discussion on policy recommendations and improving adherence is scattered throughout the introduction. Instead, create a distinct section towards the end that summarizes key points effectively.

2. Justification of Study Significance:

• The introduction provides a solid background on the importance of iron-folic acid supplementation but does not sufficiently highlight the novelty of the study in comparison to existing literature. The authors should clarify how their findings contribute to addressing gaps in current knowledge.

3. Methodological Clarity:

• The study mentions the use of a multivariable multilevel mixed-effects logistic regression model but does not provide sufficient justification for selecting this specific approach. Could the authors elaborate on why this method was chosen over alternative statistical models?

• The selection criteria for variables in the final multivariable model should be explicitly detailed. Why was a p-value threshold of 0.25 used in the bi-variable model for inclusion in the multivariable analysis?

• The manuscript states that 8460 weighted study participants were used. However, were there any missing data, and if so, how was this handled?

• The manuscript refers to a pooled analysis, but it only utilizes data from the 2022 survey. The authors should clarify whether they intended to conduct a pooled analysis over multiple years or rephrase this to reflect that only 2022 data were analyzed.

4. Interpretation of Results:

• The discussion on adherence prevalence could be expanded by comparing it with similar studies in other regions, particularly in sub-Saharan Africa.

• The study finds that women with higher education levels are more likely to adhere to iron-folic acid supplementation. What implications does this have for public health interventions aimed at less-educated women?

• The study mentions a 70% reduction in adherence among women who start antenatal care in the third trimester. The authors should discuss the practical implications of this finding, including policy recommendations to encourage earlier antenatal visits.

5. Policy and Practical Recommendations:

• While the study concludes that adherence is associated with maternal age, education, and antenatal visits, there is limited discussion on policy-level interventions. What are the specific strategies that the Kenyan health system can implement to improve adherence rates?

• The study suggests that healthcare professionals should be trained, but it would be helpful to specify the kind of training and how it could be practically implemented.

6. Sentence inconsistency:

• Correct the sentence in line 104

• Rephrase the section "Implications for policy, practice and future research". Need to improve the sentence-making standard. Especially look at lines 478-480

6. PLOS authors have the option to publish the peer review history of their article (what does this mean? ). If published, this will include your full peer review and any attached files.

**Do you want your identity to be public for this peer review?** For information about this choice, including consent withdrawal, please see our Privacy Policy .

Reviewer #1: **Yes: ** Perdomo Sandoval, Luis Albeiro

Reviewer #2: **Yes: ** First name: Habtamu Molla

Last name: Ayele

Affilation: Maternal and Child Health Directorate, Federal Ministry of Health, Addis Ababa, Ethiopia.

Reviewer #3: No

---

## [Author Response · Author response to Decision Letter 1]

26 Mar 2025

March 26, 2025

Dear Editors and Reviewers

We have submitted these authors' responses to peer-review comments and questions for a manuscript entitled: Adherence to iron-folic acid supplementation and its associated factors among pregnant women in Kenya: a multilevel data analysis of 2022 Kenyan national population-based survey.

Dear Editor, first, we would like to thank the PLOS ONE Editorial Office members, especially the chief editor, who timely assigned the competent academic editor and facilitated the progress of the review forum. Second, our special thanks go to the academic editor for his editorial contribution and for assigning the potential skilled and experienced reviewers in the field promptly. Moreover, our deepest gratitude goes to the esteemed reviewers for their constructive comments and scientific contributions that helped us to improve the quality of this manuscript. Dear academic Editor and reviewers, we have made the necessary corrections and responses to those comments raised by academic editor and reviewers point by point, page by page, and line by line sequentially. Accordingly, we have added the authors’ responses in the review forum, as well as the revised manuscript and the tracked changes to the manuscript on the web page.

With regards!

Gebrie Getu Alemu

gebryegetu27@gmail.com

University of Gondar, Ethiopia

PO. Box 196

Corresponding Author

---

## [Editor Report · Decision Letter 1]

PONE-D-24-57968R1Adherence to iron-folic acid supplementation and its associated factors among pregnant women in Kenya: a multilevel data analysis of 2022 Kenyan national population-based surveyPLOS ONE

Dear Dr. Alemu,

Thank you for submitting your manuscript to PLOS ONE. After careful consideration, we feel that it has merit but does not fully meet PLOS ONE’s publication criteria as it currently stands. Therefore, we invite you to submit a revised version of the manuscript that addresses the points raised during the review process. I have some concern in the manuscript. I enjoyed reading your manuscript which still requires revision. I have included my comments below. I suggest to rework in almost in all the section. Remove unnecessary wording, try to make it catchy with important information. Provide an opportunity to proofread to your colleagues before submission which may help to reduce unnecessary words. 

We look forward to receiving your revised manuscript.

Kind regards,

Kanchan Thapa, MPH, MPhil

Academic Editor

PLOS ONE

**Additional Editor Comments:**

Dear Authors

I suggest to make your abstract catchy, don’t mention all the categories of the age. Try to mention one or main highlight about the categories of age.

We dont need a separate research question. Therefore, please incorporate in the last paragraph of the introduction section.

Why not to separate into three different section for this?- Study setting, design and periods.

Source population- it is better to mention the number of the source population. How many women delivered in the year 2022 and provide the relevant citation. Line 218- cite the relevant reference.

Ethical approval and consent to participate- Please reorganize the section. Mention about the details how the ethical consideration was done, how informed consent was taken, how the interview was done? Please read the KDHS once again for details.

Line 216-265. Please include all the information in a table a supplementary file. Just brief about the variables you have considered for analysis in a paragraph.

Incorporate the (Data management and analysis) into statistical analysis section.

Results-

Table 1. Please take care of elements mention?

Maternal age should mention in years, right? Maternal age( Years)

Also, how to mention 45-49 (73) and 0.86? – I think it is better to incorporate into previous category.

It is important to create a subsection for- Prevalence of adherence to iron-folic acid supplementation among 350 pregnant women in Kenya. I found the result was mention in only a line. I think it is better not to mention a separate section.

Present your findings in a analytical way.

Discussion-

Line 397-404- Please highlight your findings, what is your major findings of your study. Please review relevant literature to organize your section.

You need to compare and contrast your result. Please compare and contrast your result and provide the reason why it is so? – In your example- Line 404-406, how it is similar to WHO recommendation, what does WHO talks about IFAS? Also, why it is similar to Ethiopia?

The discussion section is loosely written. Please rewrite the whole section, reviewing similar literature published in PLOS One. I found the reference are not organized as per the standard, please follow the standard guideline of PLOS One. See the full reference in the link- https://pubmed.ncbi.nlm.nih.gov/31777771/

What is SSA in the reference 12 in the text in the discussion section, I need to read the reference list to understand it. Please mention the full form whenever necessary?

Take care of all the referencing throughout the paper.

Strengths and limitations of the study-

Separate into two paragraph. What are the strengths, policy relevance, global relevance, developing countries relevancy, why it is important, what is the extra that this report explored rather than DHS report?

What are the limitations?

Conclusion- Write your conclusion based on your results, some of the information about WHO recommendation need to go to the discussion section.

I found this section is very important- Implications for policy, practice and future research. However, I suggest to limit your words and incorporate into the strengths and limitation section.

Figure 2. Is it important to mention outcome status in the figure? Please take care of your color used? Make the figure standard, display the number and percentage together. Legend should mention in standard way, why to include black background here?

I found reference are not accurately organized in reference list. Please take care while citing and referring. See reference 16- MoH Kenya. National Iron & Folic Acid Supplementation, where can we read the document?

To cite KDHS, you need to use the given reference in the report. Please read the report once again. I found the given citation for KDHS is- KNBS and ICF. 2023. Kenya Demographic and Health Survey 2022. Key Indicators Report. Nairobi, Kenya, and Rockville, Maryland, USA: KNBS and ICF.

---

## [Author Response · Author response to Decision Letter 2]

5 May 2025

May 05, 2025

Rebuttal letter

To: PLOS ONE Editorial Office

Authors’ point-by-point Responses to a Manuscript ID: PONE-D-24-57968R1

Dear Editor in Chief, Greetings.

We have submitted these authors' responses to peer-review comments and questions for a manuscript entitled: Adherence to iron-folic acid supplementation and its associated factors among pregnant women in Kenya: a multilevel data analysis of the 2022 Kenyan Demographic and health Survey

Dear Editor, first, we would like to thank the PLOS ONE Editorial Office members, especially the chief editor, who timely assigned the competent academic editor and facilitated the progress of the review forum. Second, our special thanks go to the academic editor for his editorial contribution and for assigning the potential skilled and experienced reviewers in the field promptly. Moreover, our deepest gratitude goes to the esteemed reviewers for their constructive comments and scientific contributions that helped us to improve the quality of this manuscript. Dear academic Editor, we have made the necessary corrections and responses to those additional comments raised by the editors point by point, page by page, and line by line sequentially. Accordingly, we have added the authors’ responses in the review forum, as well as the revised manuscript and the tracked changes to the manuscript on the web page.

With kind regards!

Gebrie Getu Alemu

gebryegetu27@gmail.com

University of Gondar, Ethiopia

PO. Box 196

Corresponding Author

---

## [Decision Letter · Decision Letter 2]

Adherence to iron-folic acid supplementation and its associated factors among pregnant women in Kenya: a multilevel data analysis of the 2022 Kenyan Demographic and Health Survey

PONE-D-24-57968R2

Dear Dr. Alemu,

We’re pleased to inform you that your manuscript has been judged scientifically suitable for publication and will be formally accepted for publication once it meets all outstanding technical requirements.

Kind regards,

Kanchan Thapa, MPH, MPhil

Academic Editor

PLOS ONE

Additional Editor Comments (optional):

Dear Authors

Greetings

I am happy to read the revised version of the paper and I appreciate your works which will definitely help to guide the policy making, program implementation and academia in Kenya and in the world.

Congratulations

Best

Kanchan

Reviewers' comments:

Reviewer's Responses to Questions

**Comments to the Author**

1. If the authors have adequately addressed your comments raised in a previous round of review and you feel that this manuscript is now acceptable for publication, you may indicate that here to bypass the “Comments to the Author” section, enter your conflict of interest statement in the “Confidential to Editor” section, and submit your "Accept" recommendation.

Reviewer #4: All comments have been addressed

Reviewer #5: All comments have been addressed

2. Is the manuscript technically sound, and do the data support the conclusions?

Reviewer #4: Yes

Reviewer #5: Yes

3. Has the statistical analysis been performed appropriately and rigorously? 

Reviewer #4: Yes

Reviewer #5: Yes

4. Have the authors made all data underlying the findings in their manuscript fully available?

Reviewer #4: Yes

Reviewer #5: Yes

5. Is the manuscript presented in an intelligible fashion and written in standard English?

Reviewer #4: Yes

Reviewer #5: Yes

6. Review Comments to the Author

Reviewer #4: The manuscript is technically sound and reads well. The statistical analysis is sound, and discussion is well written.

Reviewer #5: The authors have addressed all the comments made by the reviewers. The manuscript is technically sound and the conclusions are supported by data. Statsistical analysis have been performed rigorously. Data has been made available. The manuscript has been presented in an intelligible manner using standard English

7. PLOS authors have the option to publish the peer review history of their article (what does this mean? ). If published, this will include your full peer review and any attached files.

**Do you want your identity to be public for this peer review?** For information about this choice, including consent withdrawal, please see our Privacy Policy .

Reviewer #4: No

Reviewer #5: **Yes: ** Lucas Banda

---

## [Editor Report · Acceptance letter]

PONE-D-24-57968R2

PLOS ONE

Dear Dr. Alemu,

I'm pleased to inform you that your manuscript has been deemed suitable for publication in PLOS ONE. Congratulations! Your manuscript is now being handed over to our production team.

Kind regards,

on behalf of

Mr. Kanchan Thapa

Academic Editor

PLOS ONE